# Comparative assessment of the cost-effectiveness of Tuberculosis (TB) active case-finding interventions: A systematic analysis of TB REACH wave 5 projects

Isabella Gomes[1], Chaoran Dong[1], Pauline Vandewalle[2], Amera Khan[2], Jacob Creswell[2], David Dowdy[1‡], Hojoon Sohn[3‡]*

1 Department of Epidemiology, Johns Hopkins Bloomberg School of Public Health, Baltimore, Maryland, United States of America, 2 Stop TB Partnership, TB REACH Initiative, Geneva, Switzerland, 3 Department of Preventive Medicine, Seoul National University College of Medicine, Seoul, Republic of Korea

☯ These authors contributed equally to this work.
‡ DD and HS are co-senior authors on this work.
* hsohn@snu.ac.kr

**Data Availability Statement:** All relevant data are within the paper and its Supporting Information files.

## Abstract

### Purpose

Interventions that can help streamline and reduce gaps in the tuberculosis (TB) care cascade can play crucial roles in TB prevention and care, but are often operationally complex and resource intensive, given the heterogenous settings in which they are implemented. In this study, we present a comparative analysis on cost-effectiveness of TB REACH Wave 5 projects with diverse programmatic objectives to inform future decisions regarding funding, strategic adoption, and scale-up.

### Methods

We comprehensively reviewed project reports and financial statements from TB REACH Wave 5, a funding mechanism for interventions that aimed to strengthen the TB care cascade in diverse settings. Two independent reviewers abstracted cost (in 2017 US dollars) and key programmatic data, including project type (case-finding only; case-finding and linkage-to-care; or case-finding, linkage-to-care and patient support), operational setting (urban or rural), and project outputs (numbers of people with TB diagnosed, started on treatment, and successfully completing treatment). Cost-effectiveness ratios for each project were calculated as ratios of apportioned programmatic expenditures to corresponding project outputs.

### Results

Of 32 case finding and patient support projects funded through TB REACH Wave 5, 29 were included for analysis (11 case-finding only; 9 case-finding and linkage-to-care; and 9 case-finding, linkage-to-care and patient support). 21 projects (72%) were implemented in either Africa or Southeast Asia, and 19 (66%) focused on serving urban areas. Average cost-

**Funding:** TB REACH - an initiative of Stop TB Partnership – is funded by Global Affairs Canada grant number CA-3-D000920001. https://w05. international.gc.ca/projectbrowser-banqueprojets/ projectprojet/ details/d000920001 and The Bill and Melinda Gates Foundation (OPP1139029) https:// www.gatesfoundation.org/about/committed-grants/2015/11/opp1139029. The funders also provided support in the form of salaries for PV, AK, and JC. The funder had no role in study design, data collection and analysis, decision to publish, or preparation of the manuscript.

**Competing interests:** The authors have declared that no competing interests exist.

effectiveness was $184 per case diagnosed (range: $30-$10,497), $332 per diagnosis and treatment initiation ($123-$10,608), and $40 per patient treatment supported ($8-$160). Cost per case diagnosed was lower for case-finding-only projects ($132) than projects including linkage-to-care ($342) or linkage-to-care and patient support ($254), and generally increased with the corresponding country's per-capita GDP ($543 per $1000 increase, 95% confidence interval: -$53, $1138).

## Conclusion

The costs and cost-effectiveness of interventions to strengthen the TB care cascade were heterogenous, reflecting differences in context and programmatic objective. Nevertheless, many such interventions are likely to offer good value for money. Systematic collection and analysis of cost-effectiveness data can help improve comparability, monitoring, and evaluation.

## Introduction

Tuberculosis (TB) is the second leading infectious cause of morbidity and mortality worldwide, trailing after SARS-CoV-2, with an estimated 9.9 million new TB cases and 1.5 million deaths in 2020 [1]. In 2014, the World Health Organization's End TB Strategy called for a 90% reduction in TB incidence and 95% reduction in TB mortality rates by 2035 [2]. Similarly, the Stop TB Partnership's Global Plan to End TB, launched in 2019, calls for UN member states to successfully treat 40 million people with TB and provide TB preventive therapy to at least 30 million people by 2022 [3]. But despite these ambitious goals, TB incidence and mortality are falling at no more than 2–4% per year—far below the reduction needed (>10%) to achieve global targets [1].

Currently, it is estimated that about 30% of people who develop active TB every year will not be notified to public health authorities–largely reflecting underdiagnosis and undertreatment [4]. As people with TB who are missed can perpetuate transmission and suffer the adverse consequences of untreated disease (including death), it is imperative to identify these individuals early and ensure the rapid uptake of TB treatment, particularly among at-risk populations. Public health interventions such as intensified case finding (ICF), active case finding (ACF) and other approaches to improve gaps in the TB care cascade are therefore critical components of a comprehensive strategy to reduce the burden of TB worldwide [5, 6].

Since 2010, the TB REACH initiative of the Stop TB partnership (UNOPS), supported by Global Affairs Canada, USAID, and the Bill and Melinda Gates Foundation, has funded 8 waves consisting of 313 projects in 54 countries that focus on adopting innovative (both technological and process) approaches to improve TB case detection and treatment. In particular, Wave 5 –the focus of this analysis–focused on innovative approaches to case finding. These projects have made important contributions in innovating and promoting TB case finding activities in many high-burden TB countries [7, 8]. However, TB case finding and treatment projects are resource intensive, and their cost-effectiveness–both in absolute and comparative terms–remains uncertain [9, 10].

Given the substantial investment made in these projects (over 155 million USD since the inception of the initiative), it is critical to understand the relative value generated by different types of case finding interventions. In this study, we used the project database of the TB

REACH Wave 5 funded projects to comprehensively and systematically evaluate costs and cost-effectiveness across the wide range of case finding and treatment support projects supported by this initiative. In doing so, we compared cost-effectiveness ratios (CERs) [6] assessed for common outcome units that are most direct and readily calculated to measure ACF performance based on each project's scope of operations (1. TB case finding; 2. Linkage to care; and 3. TB treatment support) to rank and compare various factors influencing CERs.

## Methods

During the wave 5 funding cycle, TB REACH funded 32 case finding projects across 20 different countries with total support of 16 million USD. An additional six small grants were provided to develop tools or products. While the main focus of the funding cycle was TB case detection, the overall scope of projects funded was broad; some examples of these innovations included novel approaches to case finding (e.g. recruiting civilians as TB finders in the community and strengthening public-private mix (PPM) partnerships), scaling-up previously proven concepts (e.g., engaging community health workers in rural areas in active case finding), improving treatment referral and adherence among individuals diagnosed with TB through active case finding, and increasing awareness regarding TB infection in the community (e.g. involvement of mass media, implementation of educational programs, and community engagement).

### Screening and data extraction

We first obtained a complete set of administrative documentation–including project applications, reviews of project activity and financial reports–for each project funded during TB REACH wave 5. We then created a standardized data extraction spreadsheet, the components of which were based on a complete review of 32 selected case finding projects prior to data extraction.

Wave 5 projects that did not report relevant cost or program yield/performance data to TB REACH were excluded from the study. After consultation with TB REACH technical officers, three additional projects were excluded from the analysis. The first project (NTRL, EPHI) was a lab-based assessment of a novel transport and decontaminating reagent for TB testing, called OMNIgene® SPUTUM. The second project (Ifakara Health Institute) was intended to assess the use of Xpert Omni/Xpert MTB Ultra cartridges. However, these cartridges were unavailable during the wave 5 funding cycle; therefore, the project was not able to begin activities. The third project (AIGHD) aimed to establish TB screening in an HIV community testing project that was similarly postponed.

Two authors (IG and CD) independently performed the data extraction by reviewing all relevant documentation and data for each project. All disagreements between the two authors were resolved by discussion. If a consensus could not be reached, the two senior investigators (DD and HS) were consulted. During these meetings, the four authors re-evaluated the financial report in question and/or sought additional information by the TB REACH technical team (PV, AK, and JC), who provided further detail and clarification on data discrepancies and any project-specific interpretations (e.g. successes and/or challenges reported by each project impacting interpretation of data parameters).

For each program, we assigned a unique code (Table 1) and extracted data from the finalized financial statement, determined based on the last update date for each project's financial statement (Dec 31st, 2018).

Variables directly collected from financial statements included the characteristics of each program, the country in which the project was executed, a brief description of the program's

**Table 1. Project characteristics and description.**

| #[a] | Project Code | Project Title | Region[b] | Setting (Target Population)[c] | Project Type[d] | Country | GDP per capita (2017) | Total Expenditure | Project Description | Project State |
|---|---|---|---|---|---|---|---|---|---|---|
| A1 | HEAAI | Health Alliance International | AFR | Urban | Case finding and Other (Non-Case Finding) | Mozambique | $461 | $527,978 | Aims to improve TB linkage-to-care by scaling up diagnostic and lab connectivity technologies and creating a comprehensive national electronic MDR-TB testing database. | Scale-up |
| A2 | GOMSA | GomSACA | AFR | Rural (Internally Displaced Persons) | Case finding | Nigeria | $1,969 | $337,109 | Aims to promote TB/HIV awareness and improve case detection and linkage-to-care among Internally Displaced Persons by engaging community volunteers and organizations. | Start-up |
| A3 | CIDRZ | CIDRZ | AFR | Urban | Case finding and Treatment | Zambia | $1,535 | $722,266 | Aims to perform community mobilization via educational campaigns and TB messaging; and compare community-based versus facility-based TB screening. | Scale-up |
| A4 | SHDEP | SHDEPHA+kAHAMA | AFR | Urban (General population; Children, Female Sex Workers, Small-Scale Miners, MSM) | Case finding and Treatment | Tanzania | $1,005 | $295,736 | Aims to conduct community outreach TB case finding in the general population, focusing on children, female sex workers, small-scale miners and MSM via door-to-door sputum collection. | Start-up |
| A5 | LSTME | LSTM | AFR | Rural | Case finding and Treatment | Ethiopia | $768 | $192,504 | Aims to expand project that engages government-employed female Health Extension Workers in conducting community TB case finding in rural areas. | Start-up |
| A6 | CHEAS | Center for Health Solutions | AFR | Urban (Children) | Case finding | Kenya | $1,568 | $873,335 | Aims to build healthcare worker capacity in the management of pediatric TB (involves a pilot project of the naso-pharyngeal aspirate procedure). | Scale-up |
| A7 | GLRAN | GLRA | AFR | Urban (Mothers, HIV patients, Outpatients) | Case finding and Treatment | Nigeria | $1,969 | $164,520 | Aims to improve case detection and contact tracing in MNCH clinics, PLHIV/ART clinics and outpatient clinics; and improve access to TB diagnostic services and access to DOTS. | Start-up |

(*Continued*)

**Table 1.** (Continued)

| #[a] | Project Code | Project Title | Region[b] | Setting (Target Population)[c] | Project Type[d] | Country | GDP per capita (2017) | Total Expenditure | Project Description | Project State |
|------|--------------|---------------|-----------|-------------------------------|-----------------|---------|----------------------|-------------------|---------------------|---------------|
| A8 | LSTMN | LSTM | AFR | Urban | Case finding | Nigeria | $1,969 | $178,605 | Aims to engage proprietary patent medicine vendors in enrolling participants and notifying community healthcare workers, who then conduct at-home/on-site testing and treatment initiation. | Start-up |
| A9 | FUNDA | Fundacao Manhica | AFR | Urban | Case finding | Mozambique | $461 | $315,064 | Aims to screen TB/HIV household and social contacts, perform Xpert Ultra across samples, and follow up with chest X-rays and clinical visits for presumptive cases. | Start-up |
| A10 | IRDSA | IRD FZC / IRD South Africa | AFR | Urban (Children, Pregnancy) | Case finding | South Africa | $6,133 | $325,415 | Aims to improve TB case finding, linkage-to-care and treatment uptake among children and pregnancy TB cases. | Start-up |
| A11 | NAANK | N/a'an ku sê Foundation—Lifeline Clinic | AFR | Rural | Case finding and Treatment | Namibia | $5,647 | $51,576 | Aims to improve TB detection and reduce loss to follow up, catastrophic costs and TB mortality in health camps. | Start-up |
| A12 | GLOHI | Global Health Institute | AFR | Rural | Case finding and Treatment, and Other (Non-Case Finding) | Madagascar | $515 | $282,754 | Aims to conduct TB screening and testing in remote areas via community healthcare workers, human porters and drones. | Start-up |
| E1 | MERCY | Mercy Corps | EMR | Urban | Case finding | Pakistan | $1,465 | $295,048 | Aims to engage a provincial female health worker project to set up house-to-house TB screening and to facilitate referrals to health facilities. | Start-up |
| E2 | ACREO | ACREOD | EMR | Urban (Women) | Case finding | Afghanistan | $556 | $293,980 | Aims to improve TB awareness and TB screening programs via gender-sensitive, mobile TB screening services. | Start-up |
| E3 | BRICF | Bridge Consultants Foundation | EMR | Urban (Transgender People, Male Sex Workers) | Case finding and Treatment | Pakistan | $1,465 | $239,703 | Aims to train outreach workers in active case finding and improving linkage-to-care in transgender people and male sex workers. | Start-up |
| P1 | ASOCI | Asociacion Benefica PRISMA | PAR | Urban | Case finding and Treatment | Peru | $6,711 | $353,897 | Aims to train "TB finders" in community case finding activities and providing peer support to newly diagnosed TB patients. | Start-up |

(Continued)

**Table 1.** (*Continued*)

| #[a] | Project Code | Project Title | Region[b] | Setting (Target Population)[c] | Project Type[d] | Country | GDP per capita (2017) | Total Expenditure | Project Description | Project State |
|---|---|---|---|---|---|---|---|---|---|---|
| S1 | ICCDR | ICDDR | SEAR | Urban | Case finding | Bangladesh | $1,564 | $783,292 | Aims to expand a private-sector TB screening program, which involves conducting chest X-rays and using the revenue to subsidize the operational costs, diagnostic testing and treatment. | Scale-up |
| S2 | REACH | REACH | SEAR | Urban | Case finding | India | $1,981 | $934,125 | Aims to engage the private sector (practitioners, hospitals and pharmacies) in TB prevention and care through incentives; and to encourage the notification of missing TB patients across urban settings. | Scale-up |
| S3 | TBALI | TB Alert India | SEAR | Urban | Case finding | India | $1,981 | $170,735 | Aims to map private sector resources and establish one-stop diagnostic hubs with Xpert testing to improve case detection. | Start-up |
| S4 | ASHAK | Asha Kalp | SEAR | Rural (Indigenous populations) | Case finding and Treatment | India | $1,981 | $321,924 | Aims to strengthen community-based TB screening, sample transportation and follow up care services provided by lay health workers. | Start-up |
| S5 | INNOV | Innovators in Health | SEAR | Rural | Case finding and Treatment | India | $1,981 | $308,777 | Aims to conduct door-to-door screening in rural areas and minimize loss to follow up by supporting TB patients throughout the care cascade. | Start-up |
| S6 | BNMTN | BNMT Nepal | SEAR | Rural (High Risk populations) | Case finding and Treatment | Nepal | $911 | $534,740 | Aims to increase case notification of remote or high-risk populations via contact tracing in TB health camps and outpatient screening in district hospitals using GeneXpert. | Scale-up |
| S7 | OPASH | Operation ASHA | SEAR | Rural | Case finding and Treatment | India | $1,981 | $321,924 | Aims to improve TB case detection at non-functional medical centers in a mountainous region via area mapping, sputum collection and transport, and recruitment of samples to labs. | Start-up |
| S8 | MAPIN | MAP International | SEAR | Rural | Casefinding | Indonesia | $3,837 | $341,921 | Aims to raise TB awareness and facilitate linkage-to-care, TB treatment and follow-up care for patients in remote island communities. | Start-up |

(*Continued*)

**Table 1.** (Continued)

| #[a] | Project Code | Project Title | Region[b] | Setting (Target Population)[c] | Project Type[d] | Country | GDP per capita (2017) | Total Expenditure | Project Description | Project State |
|------|--------------|---------------|-----------|-------------------------------|-----------------|---------|-----------------------|-------------------|---------------------|---------------|
| S9 | RUMAH | Rumah Sakit Islam | SEAR | Urban (Children) | Case finding and Treatment | Indonesia | $3,837 | $188,183 | Aims to conduct pediatric TB case finding, which includes screening, contact investigation and reverse contact investigation via mobile X-rays and sputum induction. | Start-up |
| W1 | CATAC | CATA | WPR | Rural (Elderly population) | Case finding and Treatment | Cambodia | $1,386 | $425,709 | Aims to implement a mobile/roving active case finding initiative targeted towards the elderly population and to fund treatment at health facilities. | Scale-up |
| W2 | KHANA | KHANA | WPR | Urban | Case finding and Treatment | Cambodia | $1,386 | $357,965 | Aims to implement and evaluate three community-based case finding strategies. | Start-up |
| W3 | VNTPV | VNTP | WPR | Urban | Case finding and Treatment | Vietnam | $2,366 | $766,510 | Aims to conduct household and social contact investigation, door-to-door community screening, facility-based screening at hospitals, and post-exposure therapy. | Scale-up |
| W4 | FITVT | FIT | WPR | Urban | Case finding and Treatment | Vietnam | $2,366 | $137,008 | Aims to build the capacity of private sector providers to increase case notification and to integrate private sector TB treatment into national notification data. | Start-up |

a. The numbering code reflects the scale of the project and the region. The letter represents the region where the project was implemented, and the number is aligned with the ordering of number of patients diagnosed, which is taken as the benchmark of project size.

b. Region is grouped by the WHO definition: African Region (AFR), Region of the Americas (PAR), South-East Asia Region (SEAR), European Region (EUR), Eastern Mediterranean Region (EMR), and Western Pacific Region (WPR).

c. Projects are categorized into urban or rural settings based on the primary implementation environment. Targeted population is specified according to TB REACH narrative reports.

d. Projects are considered as treatment related when they include treatment initiation or/and adherence activities.

primary activities (i.e. community-based screening, scale-up of previous concepts, testing of new sample transport or drug delivery systems, etc.); location of project operations (facility-based, door-to-door screening by community health workers, etc.); detail regarding the program's target population (i.e. general population, gender-based or geographical subpopulations, etc.); screening, diagnosis and treatment services; and available technology (e.g. mobile-health tools for screening, mobile chest X-rays, Xpert MTB/RIF assay), as well as reported expenditures. For each project, all financial items reported, including total budget, income received, and cumulative expenditure, were extracted separately and reported in 2017 US dollars.

Upon full review of all of the projects included in our study, we defined five major categories for further subgroup analyses. These subgroupings included: 1. Technology Innovation, 2. Public-Private Mixed (PPM) Partnerships (or Private Sector Involvement), 3. Hard-to-reach Populations (e.g. villages, camps, geographically isolated regions), 4. Pregnant Women or

Pediatric Cases, and 5. Door-to-Door Screening. We also noted the projects that supplied or linked patients to preventive therapy and projects implementing ICF activities (Table 2); however, specific data or information to apportion costs or assess programmatic yields for preventive therapy were not explicitly reported. Therefore, cost-effectiveness ratios could not be estimated for provision of preventive therapy.

Additionally, we extracted data on project service outputs, including number of people diagnosed with any type of TB, number of people started on TB treatment (notifications), and number of patients successfully treated (Table 3, S1 and S3 Tables in S1 File).

## Data analysis

We categorized projects based on their respective WHO regions, national gross domestic product (GDP) per capita (reported in 2017 US dollars [11]), and project setting (urban versus rural/remote). Based on a review of all TB REACH wave 5 projects, we defined three types of programmatic activities—1. Case Finding Only, 2. Case Finding and Linkage-to-Care and 3. Case Finding, Linkage-to-Care and Patient Support (Box 1)—and assigned each project to one of these categories [2]. The main outcome of our analysis was the cost-effectiveness ratio (CER), calculated as the total estimated cost, assessed based on the cumulative expenditure, as reported by each project's financial statement, divided by the number of relevant service outputs (beneficiaries served). In summarizing CERs across multiple projects, we calculated an average CER across all contributing projects (i.e., total cost of all projects divided by total beneficiaries served). This is equivalent to a weighted average of each program's cost-effectiveness ratio, weighted by the number of beneficiaries.

---

### Box 1. Categories of program activities in TB REACH wave 5

**Case Finding:** Program activities aim to register the target population and screen people with symptoms of TB. Activities include population enrollment and systematic symptom screening. Screening may be conducted in community settings through door-to-door visits of households or risk groups (active case finding) or in facility settings through passive surveillance. Screening tools may use a mobile phone or tablet-based platform. Some projects also used mobile diagnostic technologies such as mobile X-ray with computer aided diagnosis (CXR-CAD) and GeneXpert machines installed in mobile vans/trucks. A select few projects also explored use of novel sample transport technologies such as drones to improve case finding. "Case finding only" projects may also provide at-risk patients with preventive therapy; however, they are not directly involved in treatment support or adherence (i.e. intensive patient follow-up).

**Linkage-to-Care:** Program activities aim to improve patients' linkage to care post diagnosis (refer patients for treatment initiation). Activities include open access tents which serve as the first stop point in the clinics for patients referred from different part of the clinic or community screening, outpatient care or hospitalization based on the severity of TB conditions.

**Patient Support:** Program activities aim to improve management of patients' TB treatment and drug adherence. For example, to minimize loss to follow-up in treatment initiation, some programs further engaged newly diagnosed TB patients via follow-up phone calls, home visits, or peer support.

---

**Table 2. Projects organized by subgroup.**

| # | Project Code | Bullet List |
|---|---|---|
| **Technology** | | |
| A1 | HEAAI | • linked technology (GeneXpert machines to GxAlert), created a DR-TB result database, piloted video conferencing and telementoring platform |
| A3 | CIDRZ | • used CAD CXR, PAD based system, and electronic registry |
| A7 | GLRAN | • used SMS for test result transmission |
| A9 | FUNDA | • used Xpert Ultra in the ACF package |
| A10 | IRDSA | • used mhealth app in case-finding |
| A12 | GLOHI | • used drones, evriMED devices (pillbox dispenser) and Open Data Kit (ODK) with tablets |
| S2 | REACH | • applied e-health to support case-finding |
| S3 | TBALI | • used ehealth to support case-finding |
| S9 | RUMAH | • used mobile phone screening software |
| W1 | CATAC | • deployed new mobile Xpert Ultra/CXR systems |
| **PPM (private sector involvement)** | | |
| A8 | LSTMN | • engaged patent medicine vendors |
| S1 | ICDDR | • organized training and network for private providers, health workers and DOTS facilities |
| S2 | REACH | • engaged private sectors in case-finding, notification, and linkage to care |
| S3 | TBALI | • targeted private provider attendees for case-finding |
| W4 | FITVT | • trained private providers for diagnosis, notification, referral, treatment and follow-up |
| **Hard-to-reach populations (villages, camps, isolated regions)** | | |
| A2 | GOMSA | • conducted screening and contact tracing among internally displaced populations in camps and child contacts |
| A12 | GLOHI | • conducted activities at village levels (using drones) |
| E1 | MERCY | • served patients in chest camps and community support groups |
| S4 | ASHAK | • CHWs conduced oral screening and sputum collection in tribal villages |
| S9 | RUMAH | • conducted activities at village level |
| W1 | CATAC | • conducted active case finding among elderly (55+) in villages |
| W2 | KHANA | • community leaders conducted snowball active case finding in villages |
| **Pregnant women/pediatric TB cases** | | |
| A1 | HEAAI | • served pediatric cases |
| A2 | GOMSA | • included pediatric cases (children under 6) |
| A3 | CIDRZ | • included children in target population |
| A4 | SHDEP | • included children in target populations |
| A6 | CHEAS | • served pediatric cases (children aged 0–14) |
| A10 | IRDSA | • served both women and pediatric cases |

**Table 2.** (Continued)

| # | Project Code | Bullet List |
|---|---|---|
| **A11** | **NAANK** | • attended to pediatric cases (testing via gastric aspirates) |
| **E2** | **ACREO** | • included pregnant women in patient population |
| **S1** | **ICDDR** | • served pediatric cases |
| **S8** | **MAPIN** | • health promoters conducted screening at schools and in households to find pediatric cases |
| **S9** | **RUMAH** | • served pediatric cases |
| **W2** | **KHANA** | • included pediatric TB cases |
| **W3** | **VNTPV** | • served pediatric cases |
| **Door-door screening** | | |
| **A3** | **CIDRZ** | • conducted door-to-door visits |
| **A4** | **SHDEP** | • conducted door-to-door screening in rural communities |
| **S5** | **INNOV** | • CHWs conducted door-to-door screening and TB diagnosis in rural areas |
| **S8** | **MAPIN** | • conducted door-door screening |
| **W3** | **VNTPV** | • conducted door-to-door verbally screening strategy |
| **Provision of preventive therapy[a]** | | |
| **A3** | **CIDRZ** | |
| **A4** | **SHDEP** | |
| **A6** | **CHEAS** | |
| **A7** | **GLRAN** | |
| **A9** | **FUNDA** | |
| **A10** | **IRDSA** | |
| **S9** | **RUMAH** | |
| **P1** | **ASOCI** | |
| **W3** | **VNTPV** | |

a. These projects indicated provision of TB preventive therapy, but did not specify how this was operationalized nor provided number of patients to whom TPT was provided.

For our primary analysis, we defined cumulative expenditure as the total cost of human resources, program activities, procurement of medical items, procurement of non-medical items, and direct program support, minus the cost of operational research (categories specified in each project's financial statement). If a project's data on cumulative expenditure was limited, we used income reports instead. Depending on each project's programmatic scope and components, we calculated CERs for each category of programmatic activities/outputs (Table 3 and S1-S4 Tables in S1 File): 1) case finding (cost per patient diagnosed), 2) linkage-to-care (cost per patient referred for treatment), and 3) patient support (cost per patient completing treatment). For projects reporting programmatic activities beyond case finding (i.e., activities for linkage to care and/or treatment adherence), we assessed cost estimates for each programmatic activity by first assessing expense records specific to each activity (e.g. Xpert cartridge costs were considered specific to case finding costs only) and then adding apportioned shared costs based on the ratios of programmatic outputs for each major activity (i.e. ratios of number of people diagnosed with TB, number initiated on TB treatment and/or number of patients successfully completing TB treatment). CERs were calculated for projects

**Table 3. Cost-effectiveness of TB REACH Wave 5 projects by project type.**

| # | Project Code | Region[a] | Setting (Target Population)[b] | Apportioned Costs | Number of Patients Diagnosed | Cost per Case Diagnosed[c] |
|---|---|---|---|---|---|---|
| **Case-finding only** | | | | | | |
| S3 | TBALI | SEAR | Urban | $170,735 | 5,765 | $30 |
| S1 | ICCDR | SEAR | Urban | $783,292 | 17,100 | $46 |
| S2 | REACH | SEAR | Urban | $934,125 | 8,675 | $108 |
| E1 | MERCY | EMR | Urban | $269,388 | 1,165 | $231 |
| A2 | GOMSA | AFR | Rural (Internally Displaced Persons) | $335,312 | 1,423 | $236 |
| E2 | ACREO | EMR | Urban (Women) | $287,080 | 626 | $459 |
| S8 | MAPIN | SEAR | Rural | $341,921 | 581 | $589 |
| A8 | LSTMN | AFR | Urban | $170,594 | 247 | $691 |
| A6 | CHEAS | AFR | Urban (Children) | $852,498 | 440 | $1,937 |
| A9 | FUNDA | AFR | Urban | $306,335 | 99 | $3,094 |
| A10 | IRDSA | AFR | Urban (Children, Pregnancy) | $325,415 | 31 | $10,497 |
| **Average cost ratio** | | | | | | $132 |
| **Case-finding & Linkage-to-Care** | | | | | | |
| S4 | ASHAK | SEAR | Rural (Indigenous populations) | $269,670 | 2,626 | $103 |
| A5 | LSTME | AFR | Rural | $167,519 | 599 | $280 |
| S9 | RUMAH | SEAR | Urban (Children) | $165,645 | 532 | $311 |
| S7 | OPASH | SEAR | Rural | $268,708 | 648 | $415 |
| A3 | CIDRZ | AFR | Urban | $432,511 | 1,030 | $420 |
| A7 | GLRAN | AFR | Urban (Mothers, HIV patients, Outpatients) | $146,241 | 334 | $438 |
| W3 | VNTPV | WPR | Urban | $715,774 | 1,400 | $511 |
| W4 | FITVT | WPR | Urban | $126,339 | 171 | $739 |
| A12 | GLOHI | AFR | Rural | $227,997 | 23 | $9,913 |
| **Average cost ratio** | | | | | | $342 |
| **Case-finding, Linkage-to-Care & Patient Support** | | | | | | |
| W1 | CATAC | WPR | Rural (Elderly population) | $393,924 | 2,801 | $141 |
| W2 | KHANA | WPR | Urban | $245,619 | 1,620 | $152 |
| S5 | INNOV | SEAR | Rural | $276,568 | 1,730 | $160 |
| A1 | HEAAI | AFR | Urban | $412,494 | 1,516 | $272 |
| A4 | SHDEP | AFR | Urban (General population; Children, Female Sex Workers, Small-Scale Miners, MSM) | $279,082 | 922 | $303 |
| E3 | BRICF | EMR | Urban (Transgender People, Male Sex Workers) | $222,230 | 625 | $356 |
| S6 | BNMTN | SEAR | Rural (High Risk populations) | $463,257 | 1,092 | $424 |
| A11 | NAANK | AFR | Rural | $49,335 | 24 | $2,056 |
| P1 | ASOCI | PAR | Urban | $303,919 | 94 | $3,233 |
| **Average cost ratio** | | | | | | $254 |
| **Average cost ratio (All Projects)** | | | | | | $184 |

a. Region is grouped by the WHO definition: African Region (AFR), Region of the Americas (PAR), South-East Asia Region (SEAR), European Region (EUR), Eastern Mediterranean Region (EMR), and Western Pacific Region (WPR).

b. Projects are categorized into urban or rural setting based on the primary implementation environments. Targeted population is specified according to TB REACH narrative reports.

c. Cost per case diagnosed is calculated as total case-finding costs divided by the estimated number of patients diagnosed.

individually, and (as described above) as weighted averages across projects conducting similar activities (e.g. projects involving case finding only). A simple linear regression was conducted to assess the association between CERs and country's per-capita GDP.

Sensitivity analyses were performed by varying the total costs, as well as each of the different service outputs, by +/-25% independently for each project to evaluate the potential sensitivity of the results to those outcomes. This was done for projects conducting case finding only activities, as well as projects conducting case finding and treatment-related activities.

## Results

Of the 29 projects included in our analysis, 11 solely focused on case finding while 18 had additional programmatic aims (nine included linkage-to-care, and nine also included patient treatment support). Most projects were implemented in the African region (n = 12, 41%) or South-East Asia region (n = 9, 31%). Ten projects (34%) were implemented in rural areas, and the other 19 focused on urban settings. In addition, 11 projects (38%) specifically targeted vulnerable populations (e.g. internally displaced persons, children, miners, female sex workers, people living with HIV, pregnant women, etc.) 8 projects (28%) focused on scalability (Table 1).

The weighted average CER was $184 (Range: $30 –$10,497; n = 29) per TB case detected across all projects (Table 3). For case-finding only projects, the average cost per case detected was $132 (Range: $30 –$10,497). For projects that included both case finding and treatment initiation, the weighted CER was $342 (Range: $103 –$9,913). Projects with additional programmatic efforts toward treatment adherence and patient support had a weighted mean cost per case detected of $254 (Range: $141 –$3,233).

For those projects that included linkage to care efforts after TB diagnosis, the average cost per patient referred for treatment was $30 (Range: $8 –$695) (S2 Table in S1 File). For projects that included treatment adherence and patient support programs, the average cost per TB patient completing the treatment was $40 (Range: $8 –$160) (S3 Table in S1 File).

Six projects were identified as projects with CERs above a $1,000-per-case-detected threshold. Two of these projects–NAANK and ASOCI–included treatment support efforts and were implemented in upper-middle income countries. Thus, only three case-finding only projects (CHEAS, IRDSA, and FUNDA) and one project with linkage-to-care (GLOHI) had an estimated cost per case diagnosed higher than the corresponding country's per-capita GDP (Tables 1 and 3). Programmatic setbacks were the potential reasons for these inflated costs and will be discussed further in the Discussion section.

CERs also varied with characteristics of the underlying setting. Projects in urban settings had lower CERs than those in rural contexts (e.g., $169 vs. $242 per case detected). Moreover, the CERs of projects in the African region were generally higher than of projects performed in Southeast Asia ($554 vs. $95 per case detected) (Fig 1). This finding reflected two data trends. First, a small number of African projects had very high CERs, and these projects had greater influence on average CER values. Second, on average, projects from Southeast Asian countries diagnosed more people with TB, thereby lowering the estimated cost per TB case diagnosed compared to projects from the African region (Table 3). Cost per case diagnosed increased with the corresponding country's per-capita GDP ($543 per $1000 increase, 95% confidence interval: -$53, $1138).

This plot illustrates the cost-effectiveness ratio (2017 US dollars per case of tuberculosis diagnosed) associated with each project, according to the gross domestic product (GDP) per capita in each corresponding country. Letters represent the geographic region in which the projects were performed, and numbers order projects from largest (1) to smallest within each region. In each panel, there was one project that was not shown because its associated cost-effectiveness ratio was exceptionally high (Panel A, project A10/IRDSA, cost per case diagnosed $10,497, GDP per capita: $6,133; Panel B, project A12/GLOHI, cost per case diagnosed $9,913, GDP per capita: $515).

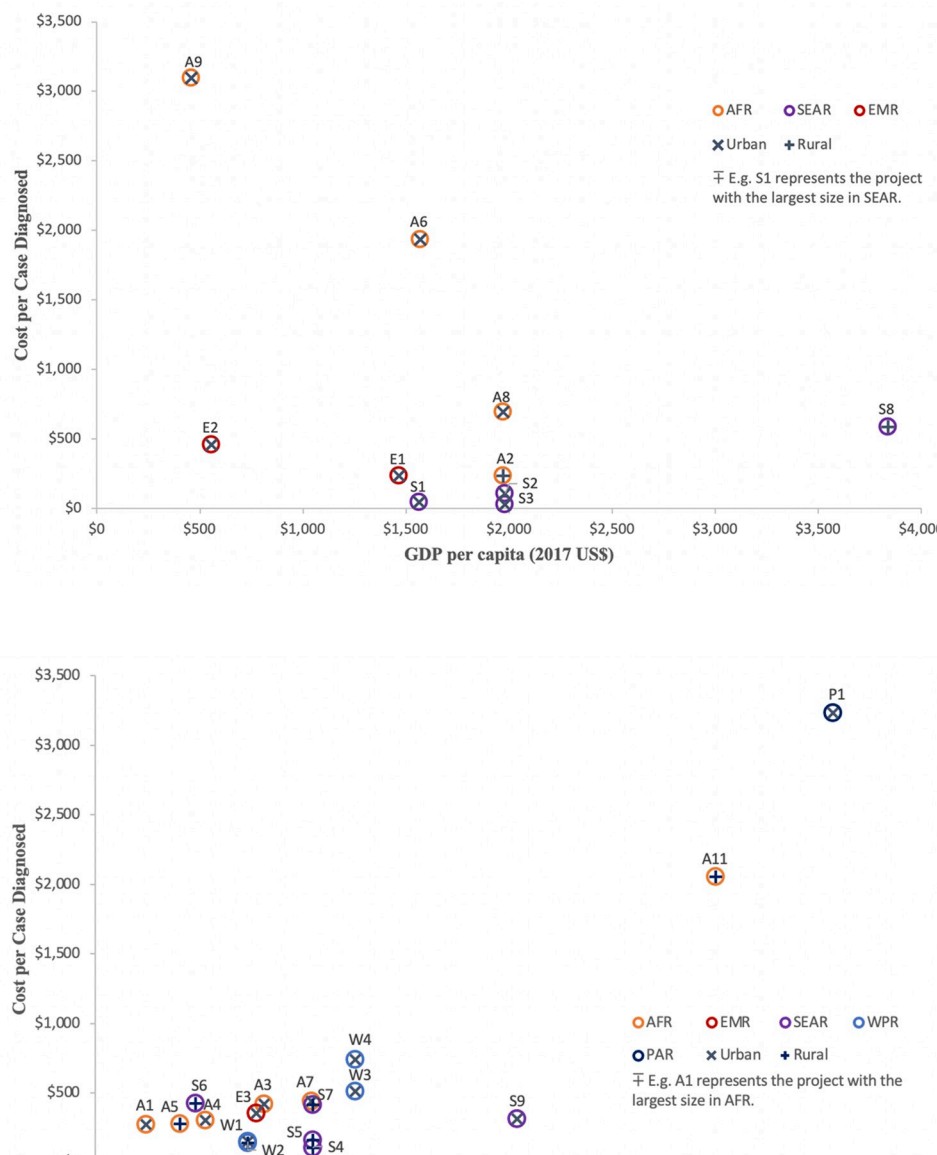

**Fig 1.** Cost-effectiveness (cost per case diagnosed) of TB REACH Wave 5 projects focused on a) case-finding only and b) case-finding and treatment support.

Subgroup analyses (S5 Table in S1 File) suggested that the average cost per case detected was highest for the projects that involved door-door screening ($361, Range: $160 –$589; n = 5), followed by projects targeting pregnant women and children ($192, Range: $46 – $10497; n = 13), projects serving hard-to-reach populations ($187, Range: $103 –$9913; n = 7), and those involving technology innovation ($169, Range: $30 –$10497; n = 10). Projects that aimed to engage the private sector through PPM partnership had an average cost of $68 per case detected (Range: $30-$739, n = 5).

In sensitivity analysis, variation in effectiveness estimates tended to have greater influence on estimated CERs than variation in costs. Varying both costs and outcomes by +/- 25% did

not affect on the characterization of projects as cost-effective (based on a cost per case diagnosed below a threshold of GDP per capita), with the sole exception of the ACREO project, which fell above this threshold when total costs increased, or case detection decreased by 25% (S1 and S3 Tables in S1 File).

## Discussion

This comparative assessment of 29 projects designed to strengthen the TB care cascade highlights the heterogeneity in cost and cost-effectiveness observed when implementing interventions with similar aims in diverse contexts. Specifically, while the majority of projects diagnosed people with TB at a cost of less than $1000 per case detected, quantitative estimates varied over 100-fold depending on the local setting, target population, specific technology or other intervention employed, methodology of implementation and assessment, and objective. These findings demonstrate the importance of considering local context and realities of implementation when evaluating cost-effectiveness and argue against making blanket statements about the cost-effectiveness of certain interventions (e.g., TB case detection). Furthermore, our specific results can be helpful to implementers and funders seeking to introduce interventions to strengthen the TB cascade of care in cost-effective fashion across a variety of diverse settings.

Despite this heterogeneity, important generalizable insights were discernible in these data. ACF projects included in our study had an overall weighted average cost per case detected under $300, below the midpoint of the corresponding opportunity-cost-based cost-effectiveness thresholds (CETs) for low-and-middle-income countries as estimated by Woods et al [12]. In most high-burden contexts, the long-term cost per disability-adjusted life year (DALY) averted has been estimated to be only modestly higher than the cost per case detected through active case-finding [6]. Thus, it is likely that these interventions would fall below country-specific CETs assessed in terms of GDP per capita in most settings with high TB burden. CETs have known limitations, and these data should not be used on their own to suggest that any specific TB case-finding intervention is cost-effective [13]. Nevertheless, these data can provide some guidance regarding the value for money of interventions to strengthen the cascade of TB diagnosis and care. Cost-effectiveness ratios were lower for projects implemented in low-income settings (where TB incidence is higher) and rural areas. Targeting hard-to-reach populations was generally not associated with an increase in cost per case detected. The cost of treatment support was generally lower than the cost to diagnose a TB case, suggesting that closing case-finding gaps requires considerably larger resource dedication than support of patients who have already been diagnosed and started on treatment. Cost-effectiveness is also more favorable in settings with higher TB incidence, as the costs of screening result in more people with TB detected and treated. These findings can help funders and policymakers prioritize project implementation and evaluation in the future. In particular, for global funding mechanisms such as TB REACH and Global Fund, interventions focused on the most disadvantaged populations (e.g., the rural poor in low-income settings) may offer optimal value for money.

Our methodology–including systematic extraction of data from standardized financial and annual reports–facilitates comparison across projects and may be useful to funding agencies when seeking to draw comparative insight on cost-effectiveness across a pre-defined set of projects. As an external benchmark of the validity of this approach, we compared our estimates with the results from a recent study by Jo et al which comprehensively evaluated the costs and cost-effectiveness of one of the projects included in this analysis (CIDRZ Zambia) [14]. Jo and colleagues used a much more detailed approach to estimate the costs of each programmatic component of this intervention and reported a cost of $435 per patient initiating treatment

[14]. Our simplified budget/finance statement-based calculation of the CER for this project was $420 per TB case diagnosed and $486 per patient initiating treatment, thus showing good agreement. The similarity of these findings may add a degree of external validation to the simple finance-report-based calculations performed here.

Four projects were identified as having higher cost-effectiveness ratios. On deeper review, these projects each confronted major operational difficulties during implementation. CHEAS experienced a major delay in implementation owing to challenges in hiring and retaining staff and underlying political unrest. IRDSA reported challenges in making household visits due to security concerns and a lower-than-expected number of patients with undiagnosed TB seeking services in the clinics. GLOHI reported that TB incidence in their region may have been over-estimated (thereby resulting in fewer patients identified) and experienced multiple technical failures in drone usage, which hindered efficiency and implementation. The FUNDA project was substantially delayed in receipt of ethical approval, leading to lower-than-expected rates of TB diagnosis. These logistical challenges speak to the unpredictable nature of scaling up health interventions in real-world settings and the resulting variation in cost-effectiveness that will likely be observed in actual implementation. These project-specific findings should not be interpreted as favoring one intervention over another–as such barriers to implementation are generally unexpected and often not related to the actual type of intervention performed.

There are several limitations to consider when interpreting our study's findings. Firstly, cost and project activity data available (level of data) for each project were highly heterogenous in that systematic assessment of activity-based costs and declassification cost data (e.g. capital assets, fixed and variable costs) across the projects were not possible. As such, we were only able to use simplified apportionment criteria to allocate total costs for major programmatic activities only: case finding, treatment initiation (linkage to care), and treatment management. This may have resulted in over-estimation of costs as certain capital assets and fixed cost items may have value beyond project activity (e.g. laboratory equipment). Additionally, several Wave 5 projects reported that provision of (e.g., to contacts of TB patients identified during ACF campaign) TB preventive therapy (TPT). However, we were unable to categorically assess costs nor performance outcomes explicitly for activities relating to TPT from data sources available to our team, including project's financial statements. Therefore, misclassification of TPT related costs, while it may be small, may have resulted in over-estimation of CERs for these projects. We also were not able to extract quantifiable data on human resource involvement that are were not compensated through the project budget (e.g. routine healthcare worker tasked to screen TB symptoms at antenatal clinic for ICF). In this regard, our results may be an under-estimate of real-world human resource costs of ACF interventions. These data limitations in our study shed light on the need to improve and standardize the types and depth of the data necessary to perform comparative in-depth empiric cost and cost-effectiveness assessment of ACF interventions [9, 14, 15]. In particular, the cost per person screened (or initiating treatment) is also useful to programs for budgeting purposes and should be a priority for future research.

Secondly, we did not use estimates of health utility such as QALYs and DALYs, as conversion from cases detected to these measures is inherently context-specific. Therefore, our study findings may not be comparable to projects with other health outcomes. However, the cost per TB case detected is arguably the most direct and readily calculated measure of ACF performance without the need to make additional strong assumptions (e.g., future trajectory of people with TB who are not detected by ACF, impact on transmission of delayed diagnosis). Earlier analyses that do make these assumptions suggest that, in high-burden countries where missed TB diagnosis is common, the cost per case detected is about 25% higher than the cost per DALY averted (when considered over a 10-year time horizon) [6]. Thus, our cost-effectiveness ratios in terms of cost per case detected are likely to be modestly higher than the

corresponding cost per DALY averted–but still well within most country-specific cost-effectiveness thresholds [12].

Thirdly, a simplified, standardized analytic approach taken in this study can be helpful for high-level comparisons on factors influencing cost-effectiveness (or costs associated programmatic performance) of interventions with similar/same objective. However, as described in our study, ACF projects are becoming more diversified to address gaps in the complex TB care cascade that are specific to the needs in the settings where these projects are implemented. Therefore, more detailed evaluation that includes projection of future benefits of key programmatic components (e.g., TPT provision to contacts of TB patients identified through ACF interventions or treatment linkages and adherence programs) is needed for each individual projects to precisely estimate the overall value and incremental cost-effectiveness of ACF interventions in each context [6, 7, 15–18].

In conclusion, this systematic comparative analysis demonstrates that the costs and cost-effectiveness of projects designed to improve the cascade of TB care are heterogeneous and context specific. This heterogeneity reflects the diversity of programmatic approaches and designs, target populations, and local settings in which these interventions are implemented. As future interventions introduce novel technology and processes, efforts to collect more detailed data that can provide insights on mechanistic, epidemiological, and operational factors influencing costs and impact of ACF interventions on the TB care cascade should be prioritized. Such data will be useful to improve our understanding of the costs and cost-effectiveness of interventions–including those interventions that fail to achieve targets for cost-effectiveness. In the meantime, projects to strengthen the TB care cascade–if implemented in locally relevant fashion–appear to offer reasonable value for money and should continue to be prioritized as part of a comprehensive approach to ending TB in high-burden settings.

## Supporting information

**S1 File.**
(DOCX)

## Author Contributions

**Conceptualization:** Isabella Gomes, Chaoran Dong, David Dowdy, Hojoon Sohn.

**Data curation:** Isabella Gomes, Chaoran Dong, Pauline Vandewalle.

**Formal analysis:** Isabella Gomes, Chaoran Dong, David Dowdy, Hojoon Sohn.

**Funding acquisition:** Amera Khan, Jacob Creswell.

**Investigation:** Isabella Gomes, Chaoran Dong, Hojoon Sohn.

**Methodology:** Isabella Gomes, Chaoran Dong, David Dowdy, Hojoon Sohn.

**Project administration:** Pauline Vandewalle, Amera Khan, Jacob Creswell.

**Resources:** Pauline Vandewalle, Amera Khan, Jacob Creswell.

**Supervision:** David Dowdy.

**Visualization:** Isabella Gomes, Chaoran Dong, Hojoon Sohn.

**Writing – original draft:** Isabella Gomes, Chaoran Dong, David Dowdy, Hojoon Sohn.

**Writing – review & editing:** Isabella Gomes, Chaoran Dong, Pauline Vandewalle, Amera Khan, Jacob Creswell, David Dowdy, Hojoon Sohn.

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
