## [Decision Letter · Decision Letter 0]

29 Jun 2021

PONE-D-21-13868

Standardized Framework for Evaluating the Cost-Effectiveness of Interventions to Improve Tuberculosis (TB) Case-Finding and Treatment Initiation

PLOS ONE

Dear Dr. Sohn,

Thank you for submitting your manuscript to PLOS ONE. After careful consideration, we feel that it has merit but does not fully meet PLOS ONE’s publication criteria as it currently stands. Therefore, we invite you to submit a revised version of the manuscript that addresses the points raised during the review process.

We look forward to receiving your revised manuscript.

Kind regards,

Martial L Ndeffo Mbah, Ph.D

Academic Editor

PLOS ONE

Additional Editor Comments:

The reviewers have raised major concerns about the quality and suitability of the manuscript, though reviewer #2 found value to the study. The paper falls short from providing a Standardized Framework for Evaluating the Cost-Effectiveness of Interventions which seems to be the main objective of the study. The objective and title of the manuscript may need to be reformulated to be more in line with the results.

Journal Requirements:

[We would like to acknowledge the TB REACH Initiative, which was funded by Global Affairs Canada, the Bill and Melinda Gates Foundation, and the United States Agency for International Development. ]

 [TB REACH - an initiative of Stop TB Partnership – is funded by Global Affairs Canada grant number CA-3-D000920001. https://w05.international.gc.ca/projectbrowser-banqueprojets/projectprojet/ details/d000920001 and The Bill and Melinda Gates Foundation (OPP1139029) https://www.gatesfoundation.org/about/committed-grants/2015/11/opp1139029. The funders also provided support in the form of salaries for PV, AK, and JC. The funder had no role in study design, data collection and analysis, decision to publish, or preparation of the manuscript.]

Reviewers' comments:

Reviewer's Responses to Questions

**Comments to the Author**

1. Is the manuscript technically sound, and do the data support the conclusions?

Reviewer #1: Partly

Reviewer #2: Yes

2. Has the statistical analysis been performed appropriately and rigorously? 

Reviewer #1: N/A

Reviewer #2: Yes

3. Have the authors made all data underlying the findings in their manuscript fully available?

Reviewer #1: No

Reviewer #2: Yes

4. Is the manuscript presented in an intelligible fashion and written in standard English?

Reviewer #1: Yes

Reviewer #2: Yes

5. Review Comments to the Author

Reviewer #1: This is an interesting topic and the need to understand the cost-effectiveness of case-finding approaches in TB is clear. However, the paper falls short of addressing the policy intent of CEA, which is to identify investment priorities across a range of options (including outside of TB). It would be a relatively simple matter to estimate DALYs averted through these policies, using an option like the freely-available DALY calculator (www.ghcearegistry.org). Exploratory analyses could be conducted to vary global disability weights for detected and treated TB vs. not, for example. Finally, comparison to GDP per capita is no longer an acceptable threshold for determining what is "cost effective", and such a standard has not been applied to cost per diagnosed case or other nontraditional metric in any event.

Reviewer #2: The authors report on the comparative cost effectiveness assessment of 29 projects, financed under the TB REACH 5 umbrella, designed to strengthen the TB care cascade. Their conclusion is that projects were heterogeneous in terms of cost and cost-effectiveness mainly due to the diverse contexts.

The methodology is appropriate; it takes advantage of the standard TB REACH format that facilitates comparison across projects.

The work, at the end, does not provide general guidance on how to design cost-effective interventions because this is basically determined by local contexts and realities of implementation. Though failing to provide a “template of evaluation” of the cost-efficacy of a project, this study has merits, as it shows that 1) projects implemented in rural and low-income settings are more cost-effective; 2) targeting hard-to-reach populations does not reduce cost-effectiveness; 3) reducing the case-finding gaps requires considerably larger resource than keeping TB cases on treatment.

Specific remarks

Abstract: the conclusion is quite generic. My suggestion would be to strengthen the concept that in the majority of projects the cascade was indeed cost effective. Just in four projects the intervention was not cost-effective due to reasons difficult to foresee and manage.

Methods: the authors report, among the limitations, a “low resolution” design (end of the discussion). It is not very clear to the reader what this refers to.

Line 247: I would expect that the average cost per case detected increase from case detection alone to case detection and treatment initiation, and case detection, treatment initiation and patient support. The authors could explain why it is not like that.

Line 267: the very large difference in costs per case detected between African and Southeast Asian countries deserves to be discussed.

Line 307: while discussing the evidence of extremely large variations in costs per TB case diagnosed, the authors speculate that these are due to local factors. However, it is also possible that the variations are due to the heterogeneity of the methodology among projects. This possibility should be discussed.

Line 383. Among the limitations, the authors include “the main outcome we measured was related to case-finding, which represents an intermediate step in the TB cascade of care. Future research should more closely evaluate interventions that focus on linkage to care, retention in care, and corresponding future health benefits.” This sentence is surprising as the authors in fact do report data on the entire cascade of care beyond the case finding step.

Line 386 onwards: I disagree on the fact that absence of reporting on preventive therapy be included among the limitation of the study. The authors might just add a short sentence in their discussion reminding about the benefits of joining these two activities

Line 403: why future programmes should be characterized by additional complexity and context dependence? This part of the sentence might be removed.

Line 407: In their conclusive sentence the authors note the importance of strengthening the TB care cascade. What is missing is a deeper discussion concerning the settings where the intervention demonstrated not to be cost-effective.

Figures: the quality of figures in the appendix is very poor

6. PLOS authors have the option to publish the peer review history of their article (what does this mean?). If published, this will include your full peer review and any attached files.

Reviewer #1: No

Reviewer #2: **Yes: **Alberto Matteelli

---

## [Author Response · Author response to Decision Letter 0]

7 Sep 2021

Editorial Comments

1. The paper falls short from providing a Standardized Framework for Evaluating the Cost-Effectiveness of Interventions which seems to be the main objective of the study. The objective and title of the manuscript may need to be reformulated to be more in line with the results.

 Thank you for this suggestion. We agree that the title of our manuscript does not

adequately match the main objective and contents of our analyses. As such, we have

revised the title to better reflect the purpose and outcome assessments made in

our work:

“Comparative Assessment of the Cost-Effectiveness of Tuberculosis Active Case

Finding Interventions: A Systematic Analysis of TB REACH Wave 5 Projects."

 Our revised version of the manuscript now reflects necessary formatting changes to adhere to the PLOS ONE’s style requirements. 

[We would like to acknowledge the TB REACH Initiative, which was funded by Global Affairs Canada, the Bill and Melinda Gates Foundation, and the United States Agency for International Development. ]

 [TB REACH - an initiative of Stop TB Partnership – is funded by Global Affairs Canada grant number CA-3-D000920001. https://w05.international.gc.ca/projectbrowser-banqueprojets/projectprojet/ details/d000920001 and The Bill and Melinda Gates Foundation (OPP1139029) https://www.gatesfoundation.org/about/committed-grants/2015/11/opp1139029. The funders also provided support in the form of salaries for PV, AK, and JC. The funder had no role in study design, data collection and analysis, decision to publish, or preparation of the manuscript.]

 Thank you for this comment. We have removed acknowledgement section in our manuscript to be aligned with journal requirements. As stated in our coverletter, we kindly request that our funding statement to be replaced with the following text in the online submission form: 

“TB REACH - an initiative of Stop TB Partnership – is funded by Global Affairs Canada grant number CA-3-D000920001 and The Bill and Melinda Gates Foundation (OPP1139029). The funders also provided support in the form of salaries for PV, AK, and JC. The funder had no role in study design, data collection and analysis, decision to publish, or preparation of the manuscript.”

Comments to the Author

Please note that all line references made in our rebuttal statement refers to the clean version of the revised manuscript (i.e. the version without track changes).

Reviewer #1: This is an interesting topic and the need to understand the cost-effectiveness of case-finding approaches in TB is clear. However, the paper falls short of addressing the policy intent of CEA, which is to identify investment priorities across a range of options (including outside of TB). It would be a relatively simple matter to estimate DALYs averted through these policies, using an option like the freely-available DALY calculator (www.ghcearegistry.org). Exploratory analyses could be conducted to vary global disability weights for detected and treated TB vs. not, for example. Finally, comparison to GDP per capita is no longer an acceptable threshold for determining what is "cost effective", and such a standard has not been applied to cost per diagnosed case or other nontraditional metric in any event.

 Thank you for your valuable comment. We agree that cost effectiveness analyses provide important evidence in identifying investment priorities across a range of

options/interventions, including outside of TB. We would like to make three arguments in response to this well-reasoned comment.

First, in the case of this manuscript, our policy goal is not to evaluate whether TB case-finding is cost-effective (relative to other interventions or to a cost-effectiveness threshold), but rather to help decision-makers understand that “TB case finding” is not a monolithic approach – and that numerous contextual factors will strongly influence the cost-effectiveness of TB case finding. Although we use cost-effectiveness ratios to make this point, we intentionally do not draw any conclusions about whether TB case-finding is (or is not) cost-effective. As such, we have revised our manuscript throughout to shift wording away from any discussion of whether ACF is cost-effective, and toward discussion that supports our primary conclusion (as stated in the Abstract): “The costs and cost-effectiveness of interventions to strengthen the TB care cascade were heterogenous, reflecting differences in context and programmatic objective. Systematic collection and analysis of cost-effectiveness data can help improve comparability, monitoring, and evaluation.”

Second, the conversion of cost per case detected and treated into a cost per DALY averted is unfortunately not straightforward. While the Global Health CEA registry DALY calculator can convert cost per TB case averted into a cost per DALY averted, the cost per TB case detected by ACF does not translate simply into a cost per case averted – because this conversion depends strongly on assumptions such as (a) what happens (in terms of costs and outcomes) to people with TB who are missed by the ACF program and (b) how much transmission is averted by earlier detection. Azman et al (reference 6) account for the second of these sets of assumptions – but the first is a particularly strong assumption and unlikely to be equivalent across different types of ACF programs in different settings. We have therefore retained our primary cost-effectiveness outcomes (as they are more readily calculated), but now include a more detailed discussion of these points on lines 384-393.

Third, we agree that comparison to GDP per capita is not an acceptable threshold. We have retained GDP per capita as a description of the various settings (e.g., to help readers understand the different economic conditions under which these ACF interventions were implemented), but we have changed the only instances of cost-effectiveness thresholds being cited to reflect country-specific cost-effectiveness thresholds (REF) rather than GDP per capita-based thresholds. 

Representative revisions made in response to this comment include:

“Understanding the comparative costs and effectiveness of similar classes of interventions implemented in different settings could support future decisions regarding funding, strategic adoption, and scale-up.” (lines 35-38)

“Of the 29 projects evaluated, 25 (86%) had an estimated cost per case detected below the midpoint of the corresponding country-specific cost-effectiveness threshold, as estimated by Woods et al (REF). In most high-burden contexts, the long-term cost per disability-adjusted life year (DALY) averted has been estimated to be only modestly higher than the cost per case detected through active case-finding [6]. Cost-effectiveness thresholds have known limitations, and these data should not be used on their own to suggest that any specific TB case-finding intervention is cost-effective.” (lines 318-325)

“However, the cost per TB case detected is arguably the most direct and readily calculated measure of ACF performance without the need to make additional strong assumptions (e.g., future trajectory of people with TB who are not detected by ACF, impact on transmission of delayed diagnosis). Earlier analyses that do make these assumptions suggest that, in high-burden countries where missed TB diagnosis is common, the cost per case detected is about 25% higher than the cost per DALY averted (when considered over a 10-year time horizon) [6]. Thus, our cost-effectiveness ratios in terms of cost per case detected are likely to be modestly higher than the corresponding cost per DALY averted – but still well within most country-specific cost-effectiveness thresholds.” (lines 384-393)

Reviewer #2: The authors report on the comparative cost effectiveness assessment of 29 projects, financed under the TB REACH 5 umbrella, designed to strengthen the TB care cascade. Their conclusion is that projects were heterogeneous in terms of cost and cost-effectiveness mainly due to the diverse contexts.

The methodology is appropriate; it takes advantage of the standard TB REACH format that facilitates comparison across projects.

The work, at the end, does not provide general guidance on how to design cost-effective interventions because this is basically determined by local contexts and realities of implementation. Though failing to provide a “template of evaluation” of the cost-efficacy of a project, this study has merits, as it shows that 1) projects implemented in rural and low-income settings are more cost-effective; 2) targeting hard-to-reach populations does not reduce cost-effectiveness; 3) reducing the case-finding gaps requires considerably larger resource than keeping TB cases on treatment.

 Thank you very much for your insightful comment. We agree that we unfortunately are not able to provide a “template of evaluation” on the cost-effectiveness of a project, but we are grateful that the Reviewer still sees the value of this work in terms of the comparative findings. We have now added (lines 404-405) a statement that future data collection efforts could be used “ultimately to develop a template for evaluating the cost-effectiveness of TB active case-finding interventions”.

Specific remarks

Abstract: the conclusion is quite generic. My suggestion would be to strengthen the concept that in the majority of projects the cascade was indeed cost effective. Just in four projects the intervention was not cost-effective due to reasons difficult to foresee and manage.

 Thank you for this comment. In keeping with comments made by Reviewer 1 above, we are hesitant to put too much weight on specific cost-effectiveness thresholds. However, we agree with the Reviewer that in most cases, projects were likely to be cost-effective. We have therefore added a statement to the conclusion (lines 52-53): “Nevertheless, many such interventions are likely to offer good value for money.”

Methods: the authors report, among the limitations, a “low resolution” design (end of the discussion). It is not very clear to the reader what this refers to.

 Thank you for this comment. Our initial intent when describing “resolution” of costing data was to discuss the value of additional details (for example, more specific unit costs, description of how costs vary with different patient volumes, etc), which could inform a more detailed comparison between projects. However, we agree with the reviewer that the term “low resolution” is potentially confusing, and we have replaced this with a call to collect “more detailed” cost data instead. Specifically, in lines 412-414, we have made revisions to provide additional details: 

“efforts to collect more detailed data that can provide insights on mechanistic, epidemiological, and operational factors influencing costs and impact of ACF interventions on the TB care cascade should be prioritized.”

Line 247: I would expect that the average cost per case detected increase from case detection alone to case detection and treatment initiation, and case detection, treatment initiation and patient support. The authors could explain why it is not like that.

We agree with the Reviewer and had this same expectation ourselves. In this study, on average, projects with patient support activities had larger operational costs, but also had a greater number of patients diagnosed (Table 3). The cost-effectiveness for projects relating to case-finding and linkage to care was also influenced by a single project (GLOHI) with a very large cost per case diagnosed. Ultimately, the cost-effectiveness of each intervention was more strongly influenced by contextual factors (such as country setting, urban/rural setting) than by whether linkage to care was offered. We have intentionally refrained from any comparison across the types of programs, for this reason. For a more appropriate comparison between case-finding and linkage to care with vs without patient support, a separate study of individual projects (that are more comparable in other aspects) would be needed.

“This finding reflected two data trends. First, a small number of African projects had very high CERs, and these projects had greater influence on average CER values. Second, on average, projects from Southeast Asian countries diagnosed more people with TB, thereby lowering the estimated cost per TB case diagnosed compared to projects from the African region (Table 3). This was due to 1) small number of African projects had very high CERs that had greater influence on average CER values and 2) on average, projects from Southeast Asian countries were able to diagnose more TB patients and this resulted in cost per TB case diagnosed to be lower in general (Table 3) compared to those implemented in the African region.” 

Line 267: the very large difference in costs per case detected between African and Southeast Asian countries deserves to be discussed.

 Thank you for this comment. We have included additional text in the result section to provide a discussion on the differences in the mean cost-effectiveness ratios calculated for projects implemented in the African region versus Southeast Asian region (lines 265-270) 

Line 307: while discussing the evidence of extremely large variations in costs per TB case diagnosed, the authors speculate that these are due to local factors. However, it is also possible that the variations are due to the heterogeneity of the methodology among projects. This possibility should be discussed.

 Thank you for this insightful comment. We have added to this line (310), “methodology of implementation and assessment” as an alternative explanation. We also highlight the possibility of different methodologies in lines 409-410: “This heterogeneity reflects the diversity of programmatic approaches and designs, target populations, and local settings in which these interventions are implemented.” 

Line 383. Among the limitations, the authors include “the main outcome we measured was related to case-finding, which represents an intermediate step in the TB cascade of care. Future research should more closely evaluate interventions that focus on linkage to care, retention in care, and corresponding future health benefits.” This sentence is surprising as the authors in fact do report data on the entire cascade of care beyond the case finding step.

 Thank you for this comment. We agree that some of the projects assessed here included linkage to care and patient support, but for purposes of including all projects (including those without such components), we used an intermediate cost-effectiveness outcome. We agree with the Reviewer that our previous wording could cause confusion among some readers. As such, we have revised this sentence (lines 398-400) to read, “Future research should also incorporate the future health benefits of treatment completion (and, where applicable, preventive therapy), not just of diagnosis and treatment initiation.”

Line 386 onwards: I disagree on the fact that absence of reporting on preventive therapy be included among the limitation of the study. The authors might just add a short sentence in their discussion reminding about the benefits of joining these two activities

 Thank you for this comment. In response, we have removed the discussion of reporting on preventive therapy as a limitation of the study, including a short reference to the value of future research incorporating the benefits of preventive therapy (shown in the response to the comment above).

Line 403: why future programmes should be characterized by additional complexity and context dependence? This part of the sentence might be removed.

 We agree and have removed this part of the sentence.

Line 407: In their conclusive sentence the authors note the importance of strengthening the TB care cascade. What is missing is a deeper discussion concerning the settings where the intervention demonstrated not to be cost-effective.

 We agree and have altered the preceding sentence (lines 415-417) to highlight this point:

“Such data will be useful to improve our understanding of the costs and cost-effectiveness of interventions – including those interventions that fail to achieve targets for cost-effectiveness”

Figures: the quality of figures in the appendix is very poor

 Thank you for this comment. We have updated our figures in the appendix with higher resolution images.

---

## [Decision Letter · Decision Letter 1]

30 Nov 2021

PONE-D-21-13868R1Comparative Assessment of the Cost-Effectiveness of Tuberculosis (TB) Active Case-Finding Interventions: A Systematic Analysis of TB REACH Wave 5 ProjectsPLOS ONE

Dear Dr. Sohn,

Thank you for submitting your manuscript to PLOS ONE. After careful consideration, we feel that it has merit but does not fully meet PLOS ONE’s publication criteria as it currently stands. Therefore, we invite you to submit a revised version of the manuscript that addresses the points raised during the review process.

We look forward to receiving your revised manuscript.

Kind regards,

Martial L Ndeffo Mbah, Ph.D

Academic Editor

PLOS ONE

Journal Requirements:

Reviewers' comments:

Reviewer's Responses to Questions

**Comments to the Author**

1. If the authors have adequately addressed your comments raised in a previous round of review and you feel that this manuscript is now acceptable for publication, you may indicate that here to bypass the “Comments to the Author” section, enter your conflict of interest statement in the “Confidential to Editor” section, and submit your "Accept" recommendation.

Reviewer #3: (No Response)

Reviewer #4: (No Response)

Reviewer #5: (No Response)

2. Is the manuscript technically sound, and do the data support the conclusions?

Reviewer #3: Yes

Reviewer #4: Yes

Reviewer #5: Yes

3. Has the statistical analysis been performed appropriately and rigorously? 

Reviewer #3: Yes

Reviewer #4: Yes

Reviewer #5: Yes

4. Have the authors made all data underlying the findings in their manuscript fully available?

Reviewer #3: Yes

Reviewer #4: Yes

Reviewer #5: Yes

5. Is the manuscript presented in an intelligible fashion and written in standard English?

Reviewer #3: Yes

Reviewer #4: Yes

Reviewer #5: Yes

6. Review Comments to the Author

Reviewer #3: This is a revision of a previously submitted manuscript by Gomes and colleagues detailing a comparative assessment of different case finding approaches employed in TB REACH Wave 5 grants.

Comments

-Abstract, Purpose: Replace the words “TB control” with “TB prevention and care” in line with guidance about reducing the use of potentially stigmatizing language in TB research.

-Abstract, Methods/Results: How was the following calculated: “…with the corresponding country’s per-capita GDP ($543 per $1000 increase, 95% confidence interval: -$53, $1138).” I see no mention of this in the main text. Please remove or report in the main text, as well as the methods used to estimate this. Apologies if I missed this.

-Introduction, line 59: Please update with 2020 data and reference TB as the second leading cause of death due to an infectious disease (trailing SARS-CoV-2).

-Line 68: I believe the same data is estimated annually in the Global TB Report (case under-ascertainment). It would be useful to reference the most recent data, in light of the ongoing pandemic.

-Introduction: Did any TB REACH project employ “intensified case finding” – such as adding TB screening to routine clinic visits? It would be good to mention this as another method of case detection. I think the authors have grouped ICF under the ACF umbrella – which is fine, but there is a bit of nuance as ICF is generally less resource intensive.

-Methods, line 167: Is there a figure supplied which details eligible projects and reasons for exclusion?

Methods, Cost Apportioning: I am curious how the authors dealt with capital costs in projects? Having reviewed some TB REACH expenditures myself, this is problematic. For example, software is purchased or a diagnostic modality is purchased, however the included participants are only a subset of the patients that would be eligible outside of an operational research project. If the diagnostic modality is expected to last beyond the project, including its full cost would overestimate costs. Further, if only a subset of patients are included, then CER may be overestimated as in practice, more patients would use the intervention, reducing the ‘cost per patient’ for fixed costs. I don’t see a mention of how this was handled in the methods and I think it needs to be addressed somewhere.

Methods, Human Resource Costs: Can the authors comment on how human resources were considered? Would routine personnel implementing an intervention as part of a TB REACH project be included in reports? For example, if a healthcare worker at an antenatal clinic is now tasked with TB symptom screening as part of the study, would their additional time/cost appear in the human resources cost? If not or if this is uncertain, it must be listed as a limitation.

Methods/Results, Efficiency of Programs: Did the authors further consider the efficiency of interventions? This is complementary to the CER. For example, it allows you to assess how many participants had to be screened to detect one case of TB and if efficiency impacted CER. If such data were not available in final reports, it should be noted. Relatedly and while outside the scope of your analysis, it would also be good to mention in discussion that cost per person screened is extremely useful to programs for budgeting purposes and is an area of further research.

Reviewer #4: This study presents a comparative analysis on cost-effectiveness of TB REACH Wave 5 projects, considering single measures of health outcome in the assessment. The analysis starts with a systematic review of projects, where the authors categorize the projects in the data extraction step. Then, they evaluate the projects by comparing their cost-effectiveness on single measures of health outcome: number people diagnosed, number of treatment initiations, number of treatment completions. The interventions used for active case finding (ACF) activities and the impact of these ACF interventions might be different in each project. So, it would be good to see a comparison between cost-effectiveness of active-case finding activities and passive, facility-based case-finding alone, since it is not intuitively clear what is cost-effective in the analyses. The cost-effectiveness ratio alone is not sufficient to determine if the benefit of an intervention is worth its cost. We would be able to say an intervention is cost-effective based on the comparison to another use of resources or some recognized standard.

Abstract: In the “Purpose” section, the main objective of the study needs to be described clearly.

Introduction: The authors mentioned about the TB REACH initiative in general, but there is no information on why Wave 5 projects was chosen for the study. Was there a specific reason for analyzing the TB Reach Wave 5 projects or does this study serve as a representative / pilot study for the comparative assessment of the cost-effectiveness of TB REACH projects? The authors might add a short sentence to clarify it in the “Introduction” section. (In “Methods”, the sentence (lines 97-106) states that “the main focus of the funding cycle was TB case detection, the overall scope of projects funded was broad”. Was that the reason of considering Wave 5 projects in this study?) The last paragraph could be improved by adding a few sentences on the overall aim of the work and commenting whether that aim was achieved.

Methods: As it can be seen from the Stop TB website (https://stoptb.org/global/awards/tbreach/wave5.asp), it was stated that there are 38 projects funded in Wave 5. But the paper says the number is 32. Why is there a difference? Also, in the paper, it was indicated that three projects were excluded from the analysis (lines 167-169). Is it excluding 3 projects from the 32 projects or the 29 projects? This could create confusion for the audience. My understanding is that the authors excluded 3 projects from the 32 projects, since all tables include 29 projects. It would be better to mention this (excluding 3 projects) before Table 1.

- Table 1 (Project characteristics and description): If regions are abbreviated as AFR, PAR, SEAR, EUR, EMR, WPR, why were they written differently in this table? (all of them are ending with O)

- Table 2 (Projects organized by subgroup): Letter subscript a was used for “Provision of preventive therapy”, but the corresponding footnote was not given at the end of the table.

Lines 115-123: What are the inclusion and exclusion criteria that two authors considered to perform the data extraction? It would be good to include them to clarify how this step was performed.

Lines 212-213: There is no Box 3 in the manuscript or in the supporting document. There is no Box 2, either.

Results: Line 244 – 245: For this part “the weighted average CER per TB case detected across all projects”, please add reference to Table S5.

Lines 252-255: Similarly, for these projects, please add reference to Table S2 and S3, respectively.

Discussion: Lines 332-335: When comparing different types of projects, and considering some of them as “more cost-effective”, did the authors also look into the “less cost-effective” projects whether they experienced problems during implementing ACF activities? Were the “four projects” (lines 359-375) excluded from the analyses when commenting based on the comparison of the cost-effectiveness ratios (lines 332-335)?

Reviewer #5: While this is a relatively simple analysis, comparative assessments of costs and outcomes across projects is sorely needed and will be of great benefit to both researchers and funding sources for guidance on prioritizing funding decisions across projects. I commend the authors for this work. My comments reflect changes that will make the paper more cohesive and highlight outcomes that would be of more interest to the reader.

General comments: In general, the paper goes back and forth between different phrases, which makes it a bit difficult to read. This includes using two different codes for each project, switching between ACF and TB case-finding, and treatment initiation vs. linkage to care. Since the paper is reviewing multiple projects across many contexts, which can be confusing for the reader, the authors should ensure that the rest of the paper is as cohesive and streamlined as possible. There are also a few results that I believe should be highlighted and expounded upon.

Methods:

1) I would be more explicit about the costing methodology, as this sounds like macro/gross costing. Would it be possible to share a blank version of the data extraction spreadsheet in the appendix? This might help the reader understand exactly what data were extracted from the financial reports.

2) Could the authors please specify the state of the projects in wave 5? Were all of the projects just starting in wave 5, or were some of them started in previous waves, and therefore only recurring costs were included in wave 5? This could potentially have an effect on CER, as start-up costs for some interventions can be quite high, and if not incorporated, can artificially lower CERs.

3) Lines 155-159: Some projects involved the use of preventive therapy but costs for preventive therapy were not explicitly reported. Does this mean that you were not able to exclude the costs of preventive therapy from these projects, and therefore those costs are included in the CER of these projects? This needs to be explained more clearly (and if it is a limitation, discussed in limitations section).

Results:

1) Lines 257 – 261: When referring to the projects with CER >$1000, it would be beneficial to mention which of these projects were case-finding only and which included linkage to care + patient support. --- Some of these programmatic setbacks are discussed in the Discussion section (lines 359-375), but it would still be a good idea to mention briefly in the Results section that these costs were inflated by programmatic setbacks and will be discussed further in the Discussion section.

2) Lines 283 – 285, Figure 1. I am finding it confusing to refer to the projects by both their code names and also a separate alphanumeric code for the figures. Would it be possible to just use one code per project throughout the manuscript?

3) Lines 287 – 294: These results may be important enough to include in the main figures rather than in the supplementary materials. Since there are a few projects with exceptionally high CER, it would be interesting to know what characteristics of those specific projects are driving those costs (if possible). I actually think this would be more important than the sensitivity analysis, which could be shortened/potentially moved to the supplementary materials.

Discussion:

1) Line 323: This seems to be the first time the “CET” acronym has been used in the text – please define this beforehand for the benefit of the layperson.

2) Line 332 and 359: Please remove the phrase in parentheses “(i.e., more cost-effective)”, as these are not ICERs and the CERs are not being compared to a common threshold. In line 359, I would also recommend removing the phrase in parentheses “(i.e,. less favorable)”, as it is a judgment call.

3) Line 365: I would include a little bit more discussion on how incidence of TB affects CER especially of case-finding interventions, as CERs will be much higher in a low incidence setting than high incidence. This may not be very obvious to the layperson, but could be very important in funding decisions. This commentary could be included around lines 335-338, where the authors mention that the cost of treatment support is lower than the cost of diagnosis, and then tied back in to the paragraph describing the four high CER projects in lines 360 -375.

4) Line 380 – “reflected the available data that was available” is repetitive. Perhaps “paucity of available data…”?

5) I know another reviewer suggested including DALYs in the analysis, but I don’t think it makes much sense with the purpose of this paper as the effectiveness outcomes are all based on programmatic outcomes, and not meant to be compared across interventions. I know another reviewer requested the consideration of the DALY, but I think discussion of the DALY takes up space in the Discussion section that could be better used to reflect on the cost drivers of the high CER interventions and the results of the subgroup analyses. I leave this to the authors’ discretion.

7. PLOS authors have the option to publish the peer review history of their article (what does this mean?). If published, this will include your full peer review and any attached files.

Reviewer #3: No

Reviewer #4: No

Reviewer #5: No

---

## [Author Response · Author response to Decision Letter 1]

22 May 2022

We have attached a full color highlighted version of our responses to the reviewer as part of the coverletter.

---

## [Decision Letter · Decision Letter 2]

20 Jun 2022

Comparative Assessment of the Cost-Effectiveness of Tuberculosis (TB) Active Case-Finding Interventions: A Systematic Analysis of TB REACH Wave 5 Projects

PONE-D-21-13868R2

Dear Dr. Sohn,

We’re pleased to inform you that your manuscript has been judged scientifically suitable for publication and will be formally accepted for publication once it meets all outstanding technical requirements.

Kind regards,

Martial L Ndeffo Mbah, Ph.D

Academic Editor

PLOS ONE

Additional Editor Comments (optional):

Reviewers' comments:

Reviewer's Responses to Questions

**Comments to the Author**

1. If the authors have adequately addressed your comments raised in a previous round of review and you feel that this manuscript is now acceptable for publication, you may indicate that here to bypass the “Comments to the Author” section, enter your conflict of interest statement in the “Confidential to Editor” section, and submit your "Accept" recommendation.

Reviewer #3: All comments have been addressed

Reviewer #5: All comments have been addressed

2. Is the manuscript technically sound, and do the data support the conclusions?

Reviewer #3: (No Response)

Reviewer #5: Yes

3. Has the statistical analysis been performed appropriately and rigorously? 

Reviewer #3: (No Response)

Reviewer #5: N/A

4. Have the authors made all data underlying the findings in their manuscript fully available?

Reviewer #3: (No Response)

Reviewer #5: Yes

5. Is the manuscript presented in an intelligible fashion and written in standard English?

Reviewer #3: (No Response)

Reviewer #5: Yes

6. Review Comments to the Author

Reviewer #3: (No Response)

Reviewer #5: Thank you for addressing all comments. This will be a useful addition to the empirical cost-effectiveness literature for TB programs.

7. PLOS authors have the option to publish the peer review history of their article (what does this mean?). If published, this will include your full peer review and any attached files.

Reviewer #3: No

Reviewer #5: No

---

## [Editor Report · Acceptance letter]

22 Jul 2022

PONE-D-21-13868R2 

Comparative Assessment of the Cost-Effectiveness of Tuberculosis (TB) Active Case-Finding Interventions: A  Systematic Analysis of TB REACH Wave 5 Projects 

Dear Dr. Sohn:

I'm pleased to inform you that your manuscript has been deemed suitable for publication in PLOS ONE. Congratulations! Your manuscript is now with our production department. 

Kind regards, 

on behalf of

Dr. Martial L Ndeffo Mbah 

Academic Editor

PLOS ONE